# Learning to Make Generalizable and Diverse Predictions for Retrosynthesis

## Abstract

We propose a new model for making generalizable and diverse retrosynthetic re-action predictions. Given a target compound, the task is to predict the likely chemical reactants to produce the target. This generative task can be framed as a sequence-to-sequence problem by using the SMILES representations of the molecules. Building on top of the popular Transformer architecture, we propose two novel pre-training methods that construct relevant auxiliary tasks (plausible reactions) for our problem. Furthermore, we incorporate a discrete latent variable model into the architecture to encourage the model to produce a diverse set of alternative predictions. On the 50k subset of reaction examples from the United States patent literature (USPTO-50k) benchmark dataset, our model greatly improves performance over the baseline, while also generating predictions that are more diverse.

## 1 Introduction

This paper proposes a novel approach for one-step retrosynthesis. This task is crucial for material and drug manufacturing (Corey & Wipke, 1969; Corey, 1991) and aims to predict which reactants are needed to generate a given target molecule as the main product. For instance, Figure 1 demonstrates that the input molecule "[N-]=[N+]=NCc1ccc(SCCl)cc1", expressed here as a SMILES string (Weininger, 1988), can be generated using reactants "CSc1ccc(CN=[N+]=[N-])cc1" and "ClCCl". For decades, this task has been solved using template-based approaches (Gelernter et al., 1990; Satoh & Funatsu, 1999). Templates encode transformation rules as regular expressions operating on SMILES strings and are typically extracted directly from the available training reactions. The primary limitation of such templates is coverage, i.e., it is possible that none of the templates applies to a test molecule. In order to better generalize to newer or broader chemical spaces, recently developed template-free approaches cast the problem as a sequence-to-sequence prediction task. These approaches were first explored by Liu et al. (2017) using LSTM models; the current state-of-the-art performance on this task uses Transformer models (Lin et al., 2019; Karpov et al., 2019).

Out-of-the-box Transformers nevertheless do not effectively generalize to rare reactions. For instance, model accuracy drops by 25% on reactions with 10 or fewer representative instances in the

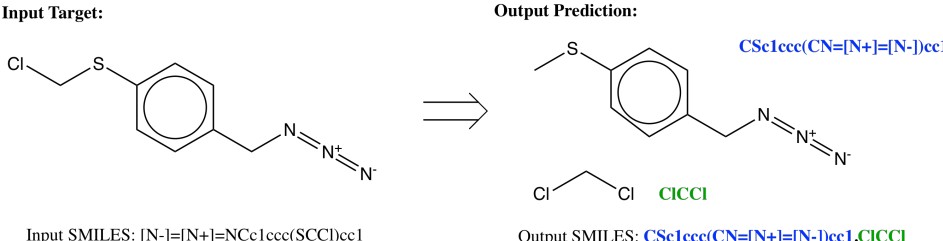

Figure 1: An example prediction task: on the left is the input target SMILES, and on the right are the output reactants SMILES. The input is a single molecule, while the output is a set of molecules separated by a period (".").

training set.[1] Another key issue is diversity. Manufacturing processes involve a number of additional criteria — such as green chemistry (having low detrimental effects on the environment). It is therefore helpful to generate a diverse collection of alternative ways of synthesizing the given target molecule. However, predicted reactions are unlikely to encompass multiple reaction classes (see Figure 2) without additional guidance. This is because the training data only provides a single reactant set for each input target, even if this is not the only valid reaction to synthesize the target.

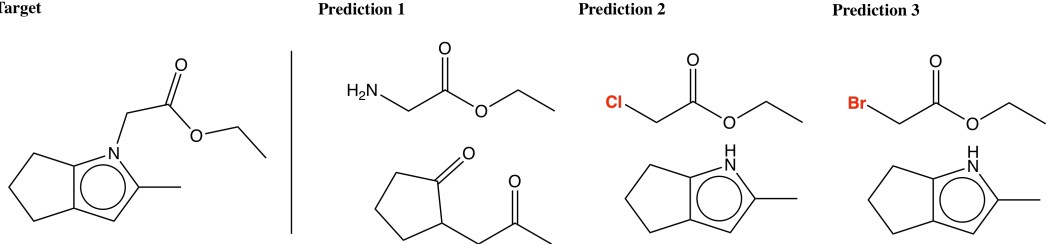

Figure 2: For the input target compound shown on the left, three possible reactant predictions are shown on the right. Prediction 1 suggestions a heterocycle formation reaction, while Predictions 2 and 3 both suggest substitution reactions. The only difference between the latter two is the halide functional group (Cl vs Br) highlighted in red. They share similar chemical properties and thus provide no additional insights for chemists.

We extend molecular Transformers to address both of these challenges. First, we propose a novel pre-training approach to drive molecular representations to better retain alternative reaction possibilities. Our approach is reminiscent of successful pre-training schemes in natural language processing (NLP) applications (Devlin et al., 2018). However, rather than using conventional token masking methods, we adopt chemically-relevant auxiliary tasks. Each training instance presents a single way to decompose a target molecule into its reactants. Here, we add alternative proxy decompositions for each target molecule by either 1) randomly removing bond types that can possibly break during reactions, or 2) transforming the target based on templates. While neither of these two auxiliary tasks are guaranteed to construct valid chemical reactions, they are closely related to the task of interest. Indeed, representations trained in this manner provide useful initializations for the actual retrosynthesis problem.

To improve the diversity of predicted reactions, we incorporate latent variables into the generation process. Specifically, we merge the Transformer architecture with a discrete mixture over reactions. The role of the latent variable is to encode distinct modes that can be related to underlying reaction classes. Even though the training data only presents one reaction for each target molecule, our model learns to associate each reaction with a latent class, and in the process covers multiple reaction classes across the training set. At test time, a diverse collection of reactions is then obtained by collecting together predictions resulting from conditioning on each latent class. Analogous mixture models have shown promise in generating diverse predictions in natural language translation tasks (He et al., 2018; Shen et al., 2019). We demonstrate similar gains in the chemical context.

We evaluate our model on the benchmark USPTO-50k dataset, and compare it against state-of-the-art template-free baselines using the Transformer model. We focus our evaluation on top-10 accuracy, because there are many equally valuable reaction transformations for each input target, though only one is presented in the data. Compared to the baseline, we achieve better performance overall, with over 13% increase in top-10 accuracy for our best model. When we create a split of the data based on different reaction templates (a task that any template-based model would fail on), we similarly observe a performance increase for our model. Additionally, we demonstrate that our model outputs exhibit significant diversity through both quantitative and human evaluations.

## 2    RELATED WORK

**Template-based Models** Traditional methods for retrosynthetic reaction prediction use template-based models. Templates, or rules, denote the exact atom and bond changes for a chemical reaction.

---

[1]See Appendix A for details on how this dataset is constructed.

Coley et al. (2018) applies these templates for a given target compound based on similar reactions in the dataset. Going one step further, Segler & Waller (2017) learns the associations between molecules and templates through a neural network. Baylon et al. (2019) uses a hierarchical network to first predict the reaction group and then the correct template for that group. However, to have the flexibility to generalize beyond extracted rules, we explore template-free generative models.

**Molecule Generation** There are two different approaches to generative tasks for molecules, demonstrated through graph and SMILES representations. The graph-generation problem has been explored in Li et al. (2018) as a node-by-node generation algorithm, but this model does not guarantee the validity of the output chemical graph. Jin et al. (2018a;b) improves upon this method using a junction-tree encoder-decoder that forces the outputs to be constrained in the valid chemical space; however, these models require complex, structured decoders. We focus on the generative task of SMILES string representations of the molecules, which has been explored in Kusner et al. (2017) and Gómez-Bombarelli et al. (2018).

**Pre-training** Pre-training methods have been shown to vastly improve the performance of Transformer models in NLP tasks without additional data supervision. Devlin et al. (2018) use a masked language modeling objective to help their model learn effective representations for downstream tasks. Similar pre-training methods on molecules have been explored by Hu et al. (2019), where they mask out atoms in molecular graphs. Meanwhile, our work does not use a masked objective, but instead creates pre-training tasks that are relevant to the retrosynthesis prediction problem.

## 3 BACKGROUND

Given an input target molecule, the task of retrosynthetic reaction prediction is to output likely reactants that can form the target product. Formally, we express a molecule as a text string via its SMILES representation, and cast our task into a sequence-to-sequence (seq2seq) prediction problem (example shown in Figure 1). For this task, the input target is always a single molecule, while the output predictions are usually a set of more than one molecule concatenated by separators ".".

To provide more intuition for this generative task, we describe some properties of SMILES strings. Each SMILES string is 1-D encoding of a 2-D molecular graph. If the predicted SMILES does not adhere to the SMILES grammar, then a valid molecular graph cannot be reconstructed. Moreover, each molecule has many equivalent SMILES representations, as a single instance of its SMILES is just a graph traversal starting at some arbitrary node. Therefore, two very different SMILES string can encode the same molecule (see Appendix B), and the model needs to be robust to the given input. One method, proposed by Schwaller et al. (2019), augments the input data with different SMILES strings of the same input target molecule.

For our model architecture, we apply a Transformer model for the seq2seq task, which has an encoder-decoder structure (Vaswani et al., 2017; Schwaller et al., 2019). The encoder maps an input sequence of tokens (from the SMILES string) to a sequence of continuous representations, which are then fed to the decoder to generate an output sequence of tokens one element at a time, auto-regressively. Once the model is trained, a beam search procedure is used at inference time to find likely output sequences.

The main building block of the Transformer architecture lies in its global self-attention layers, which are well-suited for predictions of the latent graph structure of SMILES strings. For example, two tokens that are far apart in the SMILES string could be close together topologically in the corresponding molecular graph. The global connectivity of the Transformer model allows it to better leverage this information. Additionally, since SMILES follow a rigid grammar requiring long range-dependencies, these dependencies can be more easily learned through global attention layers (see Appendix B).

## 4 APPROACH

Despite the flexible architecture of Transformer models, we recognize that there are ways to improve model generalization. Additionally, there is no inductive bias for proposing diverse outputs. We propose two techniques to enhance the base molecular Transformer model, which we describe now.

## 4.1 PRE-TRAINING

In the data, each input target molecule is associated with a single reaction transformation — though there are many equally good chemical reactions. Therefore, for each input target, we construct several new prediction examples that are chemically meaningful, and pre-train the model on these auxiliary examples. We do so without requiring additionally data, or data supervision. The two variants of our method are described in detail below, with examples shown in Figure 3.

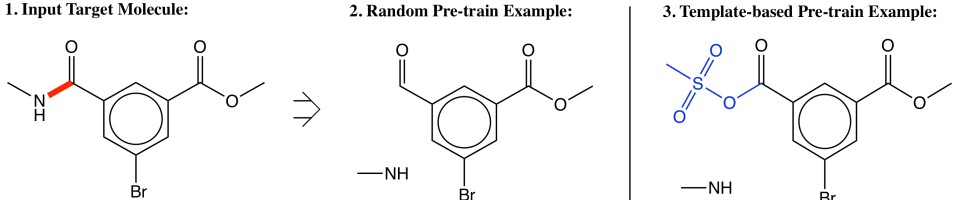

Figure 3: Input target molecule (1) with two automatically generated pre-training targets formed by breaking the bond highlighted in red. Examples (2) and (3) are generated from the random and template-based methods respectively. The only difference is that the template-based pre-training example (3) adds an additional function group to the molecule (blue).

**Random pre-training** For each input target molecule, we generate new examples by selecting a random bond to break. The types of bonds that we consider are acyclic single bonds, because these are the bonds most commonly broken in chemical reactions. As we break an acyclic bond, the input molecule is necessarily broken up into two output molecules, each being a subgraph of the input molecule. Although the examples generated by this method do not cover the entire space of chemical reactions (for instance some reactions do not break any bonds at all), these examples are easy to generate and cover a diverse range of transformations.

**Template-based pre-training** Instead of randomly breaking bonds, we can also use the templates extracted from the training data to create reaction examples. An example of a template is shown in Figure 4: each template matches a specific pattern in the input molecule, and transforms that pattern according to the template specifications. When the matched pattern is a single acyclic bond, this method will generate similar outputs as the random pre-training method, except that templates usually add additional pieces (functional groups) to the output example.

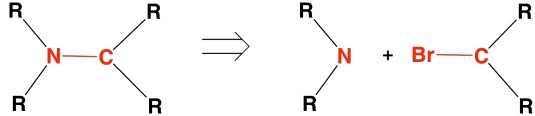

Figure 4: An example of a template, where the exact bond changes are described in red. The "C-N" bond (left) is broken and a "Br" atom is attached to the broken "C" atom (right).

As shown in Figure 3, both examples are derived from the same bond broken in the input target molecule, but for the template-based example, an additional functional group was added, matching a more realistic reaction context. On average, for a random input molecule, there are 10 different possible examples that can be extracted from the random pre-training method, while there are over 200 different possible examples that can be extracted using the template-based pre-training method. However, many of these 200 examples represent similar chemical transformations, only differing in the type of functional group added.

More broadly speaking, we can say that the template pre-training method generates more chemically valid reactions compared to the random pre-training method. However, the advantage of the random pre-training method is that it can break bonds that are not represented within the templates, thereby perhaps conferring a higher ability to generalize. As routine, the model is pre-trained on these automatically constructed auxiliary tasks, and then used as initialization to be fine-tuned on the actual retrosynthesis data.

## 4.2 MIXTURE MODEL

Next, we tackle the problem of generating diverse predictions. As mentioned earlier, the retrosynthesis problem is a one-to-many mapping since a target molecule can be formed from different types of reactions. We would like the model to produce a diverse set of predictions so that chemists can choose the most feasible and economical one in practice. However, hypotheses generated by a vanilla seq2seq model with beam search typically exemplifies low diversity with only minor differences in the suffix, see Figure 8 (Vijayakumar et al., 2016). To address this, we use a mixture seq2seq model that has shown sucess in generating diverse machine translations to generate diverse retrosynthesis reaction predictions (He et al., 2018; Shen et al., 2019).

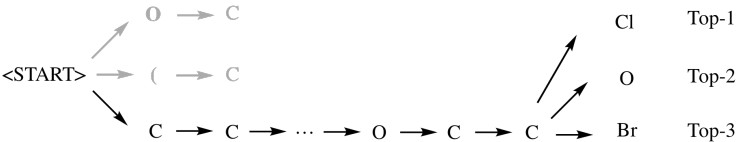

Figure 5: An example beam search; often times, the outputs of a beam search will be very similar to each other, here only differing in a single atom for the top 3 predictions.

Specifically, given a target SMILES string $x$ and reactants SMILES string $y$, a mixture model introduces a multinomial latent variable $z \in \{1, \cdots, K\}$ to capture different reaction types, and decomposes the marginal likelihood as:

$$p(y|x;\theta) = \sum_{z=1}^{K} p(y, z|x;\theta) = \sum_{z=1}^{K} p(z|x;\theta)p(y|z,x;\theta) \tag{1}$$

Here, the prior $p(z|x;\theta)$ and likelihood $p(y|z,x;\theta)$ parameterized by $\theta$ are functions to be learned.

We use a uniform prior $p(z|x;\theta) = 1/K$, which is easy to implement and works well in practice (Shen et al., 2019). For $p(y|z,x;\theta)$, we share the encoder-decoder network among mixture components, and feed the embedding of $z$ as an input to the decoder so that $y$ is conditioned on it. The increase in the parameters of our model is negligible over the baseline model.

We train the mixture model with the online hard-EM algorithm. Taking a mini-batch of training examples $\{(x^{(i)}, y^{(i)})\}_{i=1}^{m}$, we enumerate all $K$ values of $z$ and compute their loss, $-\log p(y^{(i)}|z, x^{(i)}; \theta)$. Then, for each $(x^{(i)}, y^{(i)})$, we select the value of $z$ that yields the minimum loss: $z^{(i)} = \arg\min_z -\log p(y^{(i)}|z, x^{(i)}; \theta)$, and back-propagate through it, so only one component receives gradients per example. An important detail for successfully training a mixture model is that dropout is turned off in the forward passes for latent variable selection, and turned back on at back-propagation time for gradient regularization. Otherwise even a small amount of dropout noise will corrupt the optimal value of $z$, making the selection random and the different latent components will fail to diversify (Shen et al., 2019).

The hard selection of the latent variable forces different components to specialize on different subsets of the data. As we shall later see in the experimental results, our mixture model can learn to represent different reaction types in the training data and show improved diversity over the baseline.

## 5 EXPERIMENTAL SETUP

### 5.1 DATA

The benchmark dataset we use is a subset of the open source patent database of chemical reactions (Lowe, 2012). Specifically, we use the curated 50k subset (USPTO-50k) from Liu et al. (2017), including the same data splits. Each example reaction in this dataset is labeled with one of ten reaction classes, which describes its transformation type, but we do not use this information in our experiments, similar to Karpov et al. (2019). Since we are only interested in the retrosynthesis prediction problem, the examples are processed to remove any reagent molecules (molecules that do

not contribute atoms to the reaction). The reactions are tokenized in the same manner as in Schwaller et al. (2019), with each token being a meaningful subunit of the molecule (i.e., an atom or bond).

In addition, we create a separate split of the USPTO-50k data, in which the train and test sets are split by reaction templates. Specifically, we split the data so that no example in the test set can be solved correctly with any templates extracted from training data. We use the template extraction code from Coley et al. (2017), which to the best of our knowledge, is the only publicly available template extraction code.

## 5.2 EVALUATION METRICS

**Accuracy** The evaluation of retrosynthesis is challenging, because each input target has many valid syntheses, but only one is given in the data. When the model output does not exactly match the single solution in the data, the model is not necessarily wrong, but simply giving a plausible alternative. Therefore, we focus on the top-10 accuracy for our evaluation, but present all results from our experiments. We compute the accuracies by matching the canonical SMILES strings of molecule sets. For the mixture model, we output the top 10 predictions for each latent class, and then combine those results based on likelihoods to get top 10 predictions overall.

**Diversity** To measure diversity, we provide both quantitative and human evaluations. For the former, we train a model to predict the reaction class given the input target molecule and the predicted output. We use a typical message-passing graph convolution network (Jin et al., 2017) to embed both the input and output molecules (using weight-sharing) and compute the reaction embedding as a difference of the two embeddings. This predictor is trained on the 10 reaction class labels in the USPTO-50k dataset, and achieves 99% accuracy on the test set, so we can be fairly confident in its ability to predict the reaction class in-domain.

## 5.3 BASELINES

Our main baseline is the SMILES transformer (**Base**), adapted from Schwaller et al. (2019). We run the same model as other recent works for this task (Lin et al., 2019; Karpov et al., 2019), and we build on top of the Transformer implementation from OpenNMT (Klein et al., 2017). We run ablation experiments for pre-training and different mixture models to show the impact of each approach. Random pre-training is referred to as **Pre-train (R)**, while template-based pre-training is referred to as **Pre-train (T)**. For each example, we construct up to 10 new auxiliary examples, and pre-train the model on these examples. Additionally, following Schwaller et al. (2019), we also augment the training data with variations of the input SMILES string, referred to as **Aug**. That is, for each training example, we add an extra example using a different input SMILES string, which is trained to predict the same output reactants. This helps the model learn representations robust to the permutation of the input SMILES string. In addition to our experiments, we include a template-based approach from Coley et al. (2018), and a template-free approach from Zheng et al. (2019) that adds a syntax predictor on top of the transformer model.

## 6 RESULTS

**Accuracy** The accuracy results of our model is shown in Table 1. We observe that both pre-training tasks improve over the baseline, and more so when combined with data augmentation. This shows that our pre-training tasks help the model learn the chemical reaction representational space, and are useful for the retrosynthesis prediction problem. However, interestingly, there seem to be marginal differences between the two pre-training methods. We attribute this to the fact that both pre-training methods usually generate very similar sets of examples. Previously shown in Figure 3, one of the main differences of template-based pre-training is just that it adds additional functional groups. But since these generated examples are not always chemically valid, having this extra information may not prove to be very valuable. We do note, however, that constructing additional decompositions of the input targets does actually matter for the pre-training task. We had also experimented with pre-training methods that only used variations of the input SMILES strings as pre-training output targets (because each molecule has many different SMILES representations). However, these methods did not result in the same performance gains, because these pre-training targets do not contain much useful information for the actual task.

| | Model | Top-1 | Top-2 | Top-3 | Top-4 | Top-5 | Top-10 |
|---|---|---|---|---|---|---|---|
| | Template (Coley et al., 2017) | 37.3 | - | 54.7 | - | 63.3 | 74.1 |
| | SCROP (Zheng et al., 2019) | 43.7 | - | 60.0 | - | 65.2 | 68.7 |
| Latent N = 1 | Base | 42.0 | 52.8 | 57.0 | 59.9 | 61.9 | 65.7 |
| | Aug | 44.0 | 55.3 | 60.1 | 63.0 | 65.1 | 69.0 |
| | Pre-train (R) | 43.3 | 54.6 | 59.7 | 62.4 | 64.6 | 68.7 |
| | Pre-train (T) | 43.5 | 55.6 | 61.5 | 64.8 | 67.4 | 71.3 |
| | Pre-train (R) + Aug | **(44.8)** | **57.1** | 62.6 | **65.7** | **67.7** | 71.1 |
| | Pre-train (T) + Aug | 44.5 | 56.9 | **62.7** | 65.6 | **67.7** | **71.7** |
| Latent N = 2 | Base | 42.1 | 54.4 | 60.0 | 63.1 | 64.9 | 70.3 |
| | Aug | 43.1 | 56.6 | 62.2 | 65.9 | 68.1 | 73.3 |
| | Pre-train (R) | 42.5 | 56.1 | 61.8 | 65.4 | 67.7 | 72.9 |
| | Pre-train (T) | 42.7 | 56.0 | 62.3 | 66.0 | 68.0 | 74.2 |
| | Pre-train (R) + Aug | **43.6** | **(57.7)** | 63.7 | 67.3 | 69.6 | 75.2 |
| | Pre-train (T) + Aug | 42.6 | 57.0 | **64.0** | **68.6** | **71.3** | **76.6** |
| Latent N = 5 | Base | 39.1 | 55.4 | 62.5 | 66.5 | 69.1 | 74.5 |
| | Aug | 39.7 | 56.9 | 64.1 | 68.1 | 71.1 | 77.0 |
| | Pre-train (R) | 39.7 | 55.8 | 63.5 | 67.6 | 70.1 | 76.0 |
| | Pre-train (T) | 39.9 | 54.6 | 62.9 | 68.2 | 71.2 | 77.7 |
| | Pre-train (R) + Aug | 40.2 | 56.7 | 64.9 | 69.6 | 72.4 | 78.4 |
| | Pre-train (T) + Aug | **40.5** | **56.8** | **(65.1)** | **(70.1)** | **(72.8)** | **(79.4)** |

Table 1: Accuracy metrics on the USPTO-50K dataset without reaction labels. The variations we test are data augmentation, pre-training and number of latent classes. The highest accuracy model for each different latent model is bolded, and the highest accuracy model overall is parenthesized.

Our original motivation for using a mixture model was to improve diversity, but we observe that it also leads to an increase in performance. We try $N = \{1, 2, 5\}$ for the number of discrete latent classes, and we see that more latent classes generally leads to higher accuracies. The top-1 accuracy does decrease slightly as the number of latent classes increases, but we observe much higher accuracies at top-10 (increase of 7-8%). Importantly, we note that our method of combining outputs from different latent classes is not perfect, as the likelihoods from different latent classes are not totally comparable. That is likely the cause of the decrease in top-1 accuracy; yet as we mentioned in Section 5.2, top-10 accuracies are significantly more meaningful for our problem.

| | Top-1 | Top-2 | Top-3 | Top-4 | Top-5 | Top-10 |
|---|---|---|---|---|---|---|
| Base Model | 4.3 | 8.7 | 11.9 | 14.7 | 16.6 | 20.6 |
| Best Mixture Model | 5.5 | 9.2 | 12.6 | 15.4 | 17.6 | 26.6 |

Table 2: Prediction accuracies when tested on our template split of the USPTO dataset, for which any template-based model would get 0% accuracy on the test set. We see that our template-free methods can still generalize to this test set.

Next, we show our results on a different split of the data, which is more challenging to generalize. Using the dataset split on templates described in Section 5.1, we explore the performance of our best mixture model with pre-training compared to the baseline with no pre-training. As mentioned earlier, template-free models confer advantages over template-based models, as template-based models lack the ability to generalize outside of extracted rules. For this dataset, any template-based model would necessarily achieve 0% accuracy based on construction. Table 2 compares the performance of the different models and we see that, although the task is challenging, we can attain substantial accuracy of 26.6% at top-10 compared to 20.6% of the baseline. Even on this difficult task, we show that our model offers generalizability on the test set.

|  | Unique Reactions |
| --- | --- |
| Base Model | 2.66 |
| **Mixture Model** | **3.32** |

|  | Human Diversity |
| --- | --- |
| Base Model | 21 |
| **Mixture Model** | **43** |
| Neither | 36 |

Table 3: The left table shows the comparison of number of unique reactions predicted by the base model vs. the mixture model (holding other factors constant). The right table shows human evaluation metrics, in which a human was asked to rate whether the outputs of the base model or the mixture model was more diverse, or neither.

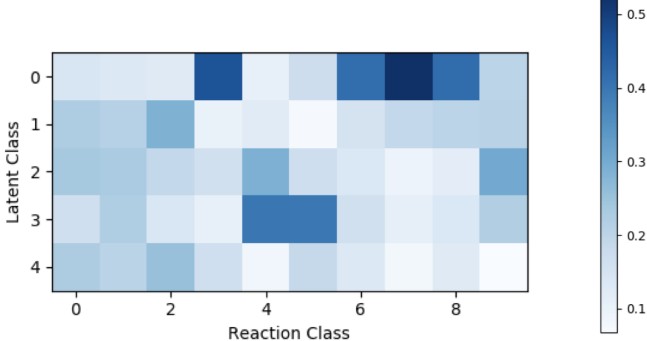

Figure 6: Heat map plotting the frequency that each latent class generates a specific reaction class. Here, we use a mixture model with 5 latent classes, and there are 10 reaction classes of interest for this data set. The reaction classes are determined by a pre-trained model described in Section 5.2

**Diversity** We now look at evaluations of diversity for our model. Using the reaction class model described in Section 5.2, we predict the reaction class for every output of our models. Then, we compute the average number of unique reaction classes, holding all other factors constant besides varying the number of latent classes (results shown in Table 3). The number of unique reaction classes is 3.32 for the mixture model compared to the 2.66 for the base model, suggesting that the mixture model predicts a more diverse cast of outputs.

The diversity of the predictions can also be examined from an interpretability standpoint for the latent classes of the mixture model. Using the reaction class model, we take the 10 top predictions from each latent class, and count the number of occurrences for each reaction class. Normalizing across reaction classes, we can see from Figure 6 that each latent class learns to predict a different distribution of reaction classes.

We also supplement our diversity results with human evaluation. To make the problem tractable for a human chemist, we randomly select 100 different reactions from the test set and present the top 5 predicted outputs from both the base and mixture model, where the the task is to determine diversity based on number of different types of reactions. The human chemist is asked to choose which of the two output sets is more diverse, or neither if the two sets do not differ in diversity (see Appendix C). For this task, the human chemist chose the mixture model more than twice as often as the base model (43 times vs 21), see Table 3. Although not perfect, these results exemplify that our model does generate more diverse outputs than the baseline.

## 7 CONCLUSION

We explored the problem of making one-step retrosynthesis reaction predictions, dealing with the issues of generalizability and making diverse predictions. Through pre-training and use of mixture models, we show that our model beats state-of-the-art methods in terms of accuracy and generates more diverse predictions. Even on a challenging task, for which any template-based models would fail, our model still is able to generalize to the test set.

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

## A  RARE REACTIONS

To compute a subset of the data with only rare reactions, we extracted all the templates from the entire USPTO-50k dataset, and selected the templates that occurs at most 10 times. The reactions in the test set that have these templates constitute the rare reaction subset, which is around 400 examples. The results for this rare reaction subset can be found in Table 4. From this table, we can see that the top-1 accuracy for the baseline model is only 18.6% which is roughly 25% drop from the 42% in Table 1. We also mention that our new models improve over this baseline, showing more generalizability.

|  | Top-1 | Top-2 | Top-3 | Top-4 | Top-5 | Top-10 |
|---|---|---|---|---|---|---|
| Base Model | 18.6 | 23.9 | 26.1 | 26.4 | 28.9 | 31.9 |
| Pre-trained Model | 19.8 | 26.4 | 29.9 | 33.4 | 35.4 | 37.7 |
| Mixture Model | 16.3 | 24.1 | 28.6 | 30.7 | 32.9 | 39.9 |

Table 4: Prediction accuracies on the rare reaction test subset.

## B  SMILES REPRESENTATIONS

Each molecule has many different SMILES representation, because each different SMILES string is just a different graph traversal over the molecule (see Figure 7). Although there is often some canonical SMILES string which is consistent, it is still completely arbitrary.

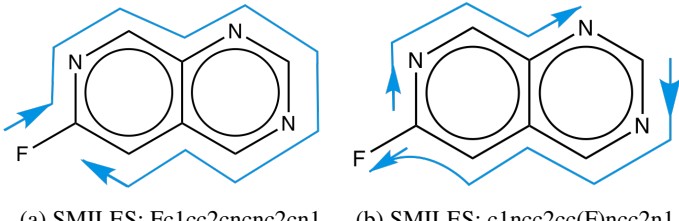

(a) SMILES: Fc1cc2cncnc2cn1    (b) SMILES: c1ncc2cc(F)ncc2n1

Figure 7: A single molecule has many different SMILES representations. On the left (a) is the canonical SMILES string, and on the right (b) is another SMILES string representing in the same molecule.

Additionally, because SMILES is a 1-D encoding of the 2-D molecular graph, two atoms that are close in the graph may be far apart in the SMILES string, shown in Figure 8. To correctly decode a SMILES string, the decoder has to be aware of long-range dependencies. For instance, numbers in the SMILES string indicate the start and end of a cycle. The decoder has to close all cycles that it starts, and at the right position, or else the output SMILES will be invalid.

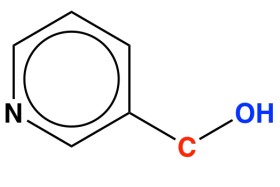

SMILES: C(c1cnccc1)O

Figure 8: The carbon atom (red) and the oxygen atom (blue) are neighbors on the molecular graph. However, in the SMILES string, they are far apart.

## C    HUMAN EVALUATION ON DIVERSITY

For human evaluation of diversity, we asked a senior (5+ years) PhD chemistry student to compare the outputs of the base model versus the mixture model. The human is given the top 5 outputs of each model and asked to rate which set is more diverse by comparing the number of reactions that differ in reaction type or location of reaction on molecule. Reactions that used slightly different precursors were considered identical, and therefore does not contribute to diversity (for example, protection reactions with different protection groups are considered as one type). Lastly, the evaluation was done with the correctness of the overall reaction in mind.

