# OpenReview forum: "Learning to Make Generalizable and Diverse Predictions for Retrosynthesis"
_ICLR.cc/2020/Conference — Reject_

### Official Review · AnonReviewer3 · 2019-10-23
**Official Blind Review #3**

**Rating:** 6

**Review:**

This paper is very well-written and combines the state-of-the-art NLP model and the domain knowledge in retrosynthetic reaction predictions.  The authors propose pre-training models to help improve the model's generation to rare reactions. In addition, a discrete latent variable model is used in the model to encourage the model to produce a diverse set of alternative predictions.  The experiments in the paper also show the effectiveness of the two main contributions.

The main contribution of this paper is to apply the state-of-the-art Transformer model and other techniques in NLP to address the specific issues in retrosynthesis. Both the pre-training model and the mixture model are combining the specific domain knowledge to improve the generalization and diversity of retrosynthesis. There is not a huge algorithm novelty for the methods proposed in this paper, but they can well address the domain issues and improve the performance.

My only concern is that the baseline model compared in the paper is Schwaller's work. I am not pretty sure if this baseline achieves state-of-the-art performance. It would be very interesting to see more comparisons with other state-of-the-art
 work.




**Experience Assessment:**

I do not know much about this area.

**Review Assessment: Checking Correctness Of Derivations And Theory:**

I assessed the sensibility of the derivations and theory.

**Review Assessment: Checking Correctness Of Experiments:**

I carefully checked the experiments.

**Review Assessment: Thoroughness In Paper Reading:**

I read the paper thoroughly.

---

> ### Author Response · Authors · 2019-11-10
> **Response to Reviewer #3**
>
> We thank the reviewer for the feedback and comments.
>
> For baselines, the vanilla transformer model from Schwaller et al. (2019) is the state-of-the-art model for retrosynthesis. Previous work using LSTM models (Liu et al., 2017) have been vastly outperformed by these transformer models. We have added the recent work of Zheng et al. (2019) in Table 1. While their method is orthogonal to our techniques, our model outperforms theirs by a substantial margin, especially for top-10 accuracy.
>
> References:
>
> (Schwaller et al., 2019): Molecular Transformer for Chemical Reaction Prediction and Uncertainty Estimation
> (Liu et al., 2017): Retrosynthetic Reaction Prediction Using Neural Sequence-to-Sequence Models
> (Zeng et al., 2019): Predicting Retrosynthetic Reaction using Self-Corrected Transformer Neural Networks

---

### Official Review · AnonReviewer1 · 2019-10-27
**Official Blind Review #1**

**Rating:** 6

**Review:**

Given a target compound, the authors suggest a method to predict likely chemical reactants to produce the target. The authors provide a transformer based model to predict the reactants. Existing methods do not generalize well for rare reactions and the training data has only one reactant set for each target even though that may not be the only way to synthesize the compound. To solve this problem, the authors use a pretraining method similar to BERT. Instead of just using token masking, they provide alternate proxy decompositions for a target molecule by randomly removing bond types that are likely to break and by transforming the target based on known templates.

This is a well written paper with good baselines.They use multiple techniques that are both domain specific (data augmentation) as well as methods from NLP adapted for this task. The experiments are carefully designed and show that both pretraining and data augmentation helps. Overall, I think the community will benefit from this work.

**Experience Assessment:**

I have published one or two papers in this area.

**Review Assessment: Checking Correctness Of Derivations And Theory:**

N/A

**Review Assessment: Checking Correctness Of Experiments:**

I carefully checked the experiments.

**Review Assessment: Thoroughness In Paper Reading:**

I read the paper at least twice and used my best judgement in assessing the paper.

---

> ### Author Response · Authors · 2019-11-10
> **Response to Reviewer #1**
>
> We thank the reviewer for the positive feedback and comments. If the reviewer has any questions or additional comments, please let us know.

---

### Official Review · AnonReviewer2 · 2019-10-27
**Official Blind Review #2**

**Rating:** 1

**Review:**

The authors present an approach to improve performance for retro-synthesis of chemical targets in a seq2seq setting using transformers. The authors have encouraging results and the paper was fairly easy to read and follow. However there are a variety of concerns that the authors need to address:

The technical contributions in this paper are somewhat thin. The main contributions are data augmentation techniques, pre-training and a mixture model that seems to improve performance on the USPTO-50K dataset. The novelty is quite low and it’s not clear if this will transfer to another domain. The impact is also low as the pre-training techniques using bond breaking and template-based are specific to this problem task. Additionally, mixture models for encouraging diversity is a simple instance of ensembling.

Using deep learning to this application area is also not novel. This paper largely builds upon previous work with Transformers from Schwaller et. al and Karpov et. al.

Other clarifications/issues:
 - The experimental results are based only on the USPTO dataset. It’s unclear how significant the results are. The authors can consider using diverse datasets or applying their techniques to another application domain to bolster their claims.
 - Table 3, lists the average number of unique reactions classes. The authors say “ … we predict the reaction class for every output of our models… “ . It’s not clear how it makes sense to calculate diversity when there’s no ground truth available for determining if the predicted output is a valid synthesis for the target. To say this another way, what good is diversity if the prediction is incorrect?
 - Table 3, lists human eval results. The details here seem quite vague. How does a human determine something to be more diverse? What is the rubric they use? How qualified is the human in being able to judge this task?
 - Figure 6 does not have a color scale.


**Experience Assessment:**

I have published in this field for several years.

**Review Assessment: Checking Correctness Of Derivations And Theory:**

I assessed the sensibility of the derivations and theory.

**Review Assessment: Checking Correctness Of Experiments:**

I carefully checked the experiments.

**Review Assessment: Thoroughness In Paper Reading:**

I read the paper at least twice and used my best judgement in assessing the paper.

---

> ### Author Response · Authors · 2019-11-10
> **Response to Reviewer #2**
>
> We thank the reviewer for the feedback and comments. We address each of them in turn:
>
> - "Novelty and domain-specificity of pre-training methods"
>
> Conventional pre-training methods use a masked language modeling (MLM) objective (Devlin et., 2018), but this kind of objective does not work for our problem. When we try a pre-training method that only masks tokens and decode the target molecule, we do not see the same performance gains.
>
> Here are some numbers after running new experiments using masked pre-training:
> No pre-training: 		                 Top-1: 42.0, Top-5: 57.0, Top-10: 65.7
> (new) Masked pre-training: 	 Top-1: 38.7, Top-5: 57.9, Top-10: 61.6
> Random pre-training: 	  	 Top-1: 43.3, Top-5: 60.1, Top-10: 69.0
> Template-based pre-training:    Top-1: 43.5, Top-5: 61.5, Top-10: 71.3
>
> While these masking objectives have worked well in NLP and vision tasks, we see that it does not work for our chemistry-domain application. In fact, we see a performance drop, probably because this pre-trained model provides a poor initialization for the task.
>
> Chemistry is a domain that involves a lot of small data problems, for which developing effective pre-training and transfer learning techniques is critically important. Therefore, we think that our effective pre-training methods are an important contribution and hope they will inspire more chemically relevant pre-training tasks for other applications.
>
> - "mixture models for encouraging diversity is a simple instance of ensembling.”
>
> We argue that our mixture model is more than a simple instance of ensembling. The mixture model encourages different latent classes to learn different reaction types and thus leads to diverse predictions. Ensembling, however, does not have this kind of inductive bias. Furthermore, an ensemble of K models would have K times more parameters, but our mixture model only requires K embeddings of the latent variable, which is a negligible amount of additional parameters.
>
> To see that the ensemble would not produce as diverse results as the mixture model, we ran our model 5 times with different random seeds, and combined the outputs based on their predicted likelihoods to produce the top 10 predictions (mirroring the setting of mixture model and results presented in Table 3).
>
> Using the reaction class predictor (we recognize that this is an imperfect metric, but is nonetheless a good proxy), we find that a single model predicts, on average, 2.7 unique reactions. The ensemble predicts 3.02 unique reactions. Our mixture model predicts 3.32 unique reactions, which is much better than that of the ensemble. While the mixture model is a simple idea, it works well to learn to make diverse predictions without additional supervision.
>
> - “Using deep learning to this application area is also not novel”
>
> We take the application of deep learning to retrosynthesis in novel directions. First, we note that diversity of predictions for this task is very important, because it is much more helpful for chemists if they have a diverse range of useful predictions. Diversity for this task has not been explored in literature and other models that focus on accuracy of their models do not capture this important facet of the problem. Second, as mentioned earlier, conventional pre-training methods are not effective for this task, so we think that our pre-training methods are a novel contribution.
>
> -“The experimental results are based only on the USPTO dataset.”
>
> The USPTO dataset is the benchmark and the only publicly available dataset used in previous work on retrosynthesis (Schwaller et al., 2019; Karpov et al., 2018, Liu et al., 2017). One work, Lee et al. (2019), did experiments on Pfizer electronic lab notebooks, but these datasets are not publicly available.
>
> - “What good is diversity if the prediction is incorrect?”
>
> We agree with the reviewer that diversity matters only when the predictions are correct. We note that our mixture models does have high accuracy, which is a good indicator that the model generally produces correct reaction outputs. We also supplement our diversity metrics with human evaluations that measure diversity only on correct predictions, see below.
>
> - “How does a human determine something to be more diverse? What is the rubric they use? How qualified is the human in being able to judge this task?”
>
> The human evaluations were done by a senior (5+ years) chemistry PhD student. They judged diversity based on the number of different reaction types and location of the reaction on the input target. The same type of reaction pathway that uses slightly different precursors is considered as identical, and this evaluation was done taking the correctness of the reaction into account. This information has been added to the appendix.
>
> - “Figure 6 does not have a color scale.”
>
> We just added it in the revision. Thank you for pointing this out.

---

> > ### Author Response · Authors · 2019-11-10
> > **(continued)**
> >
> > References:
> >
> > (Devlin et al., 2018): BERT: Pre-training of Deep Bidirectional Transformers for Language Understanding
> > (Schwaller et al., 2019): Molecular Transformer for Chemical Reaction Prediction and Uncertainty Estimation
> > (Karpov et al., 2018): A Transformer Model for Retrosynthesis
> > (Liu et al., 2017): Retrosynthetic Reaction Prediction Using Neural Sequence-to-Sequence Models
> > (Lee et al., 2019): Molecular Transformer unifies reaction prediction and retrosynthesis across pharma chemical space

---

### Decision · Program_Chairs · 2019-12-19

**Decision:**

Reject

**Comment:**

The authors present a new approach to improve performance for retro-synthesis using a seq2seq model, achieving significant improvement over the baseline. There are a number of lingering questions regarding the significance and impact of this work. Hence, my recommendation is to reject.